# Challenges of Anti-Mesothelin CAR-T-Cell Therapy

**DOI:** 10.3390/cancers15051357

**Published:** 2023-02-21

**Authors:** Xuejia Zhai, Ling Mao, Min Wu, Jie Liu, Shicang Yu

**Affiliations:** 1Department of Stem Cell and Regenerative Medicine, Institute of Pathology and Southwest Cancer Center, Southwest Hospital, Third Military Medical University (Army Medical University), Chongqing 400038, China; 2Key Laboratory of Cancer Immunopathology, Ministry of Education, Chongqing 400038, China; 3International Joint Research Center for Precision Biotherapy, Ministry of Science and Technology, Chongqing 400038, China; 4Jinfeng Laboratory, Chongqing 401329, China

**Keywords:** chimeric antigen receptor T cells (CAR-T cells), mesothelin, clinical trial, solid tumor, immunotherapy

## Abstract

**Simple Summary:**

In recent years, chimeric antigen receptor (CAR)-T-cell therapy has achieved good results in hematological malignancies. Clinical trials on anti-MSLN CAR-T cells have shown that they have a high safety profile but limited efficacy. This article reviews the clinical research status, obstacles, progress and challenges of anti-MSLN CAR-T-cell therapy and summarizes the relevant strategies to improve the efficacy and safety of anti-MSLN CAR-T-cell therapy.

**Abstract:**

Chimeric antigen receptor (CAR)-T-cell therapy is a kind of adoptive T-cell therapy (ACT) that has developed rapidly in recent years. Mesothelin (MSLN) is a tumor-associated antigen (TAA) that is highly expressed in various solid tumors and is an important target antigen for the development of new immunotherapies for solid tumors. This article reviews the clinical research status, obstacles, advancements and challenges of anti-MSLN CAR-T-cell therapy. Clinical trials on anti-MSLN CAR-T cells show that they have a high safety profile but limited efficacy. At present, local administration and introduction of new modifications are being used to enhance proliferation and persistence and to improve the efficacy and safety of anti-MSLN CAR-T cells. A number of clinical and basic studies have shown that the curative effect of combining this therapy with standard therapy is significantly better than that of monotherapy.

## 1. Introduction

The chimeric antigen receptor (CAR)-T-cell technique is a kind of cancer immunotherapy that has attracted much attention in recent years. It has achieved good curative effects in hematological malignancies, but many obstacles remain in the treatment of solid tumors. Mesothelin (MSLN) is a tumor-associated antigen (TAA) that is usually expressed only on the mesothelial surface of the body, but is significantly overexpressed in most solid tumors. This article reviews the clinical research status, difficulties and challenges of anti-MSLN CAR-T cells to provide new ideas for their application in the treatment of solid tumors.

### 1.1. CAR-T Cells

CAR is a protein composed of three parts: an extracellular antigen-binding domain, an intracellular signaling domain and a hinge region. Extracellular single-chain variable fragment (scFv) antibodies that specifically identify the surface antigen of cancer cells make up the extracellular antigen-binding domain. The target-binding domain of the CAR is composed of changeable heavy and light chains that are joined together by adaptable peptide linkers in the scFv fragment. The hinge region connects the scFv fragment to intracellular components. Through its glycine and serine sequences, the linker residue’s hydrophilicity improves flexibility, while the intermittent glutamine and lysine sequences improve solubility. The signal transduction components of T cells or natural killer cells, such as 4-1BB and CD28, which transduce extracellular binding signals to start downstream cascade stimulation signals are often the source of intracellular signaling domains. The immunoreceptor tyrosine-based activation motif (ITAM) found in the cytoplasmic CD3 domain is required for T-cell activation [1,2] (Figure 1). Four generations of CARs are currently undergoing experimental and clinical research. The difference lies in their intracellular signaling domains linked to the scFv receptor. Theoretically, third-generation CAR-T cells should have more activation and killing capacities than second-generation CAR-T cells. In addition, because of the heterogeneity of tumor cells, some tumor cells do not have antigens that can be specifically recognized by T cells and cannot be recognized and cleared by traditional CAR-T cells. This problem may be solved by fourth-generation CAR-T cells, which can recruit immune cells other than T cells to the tumor area [1,2,3] (Figure 2).

To exert antitumor effects in vivo, CAR-T cells must be persistent, proliferative and able to infiltrate tumor tissue. In general, antigens and costimulatory signals in major histocompatibility complex (MHC)-dependent complexes, which involve the recruitment of T-cell surface CD28 and the costimulatory molecules CD80 or CD86 on antigen-presenting cells (APCs), are required for T-cell-mediated immune responses. CAR-T cells recognize specific tumor antigens independent of MHC molecule restriction to perform their antitumor functions. Once CAR binds specifically to TAA, ITAM phosphorylation activates CAR-T cells and induces cytokine secretion, CAR-T-cell proliferation and cytotoxicity. CAR-T cells exert cytotoxic effects by secreting perforin and granzymes and activating death receptor signaling through Fas/Fas ligand (FasL) or TNF/TNF-α [4,5].

### 1.2. MSLN

MSLN is a glycosylated phosphatidylinositol-anchored protein that is usually expressed in small amounts on the surface of mesothelial cells in the pleura, pericardium, peritoneum and sheath (in males). However, it has been found that MSLN is overexpressed in a variety of cancers [6,7], including malignant mesothelioma [7,8,9,10,11], ovarian cancer [8,11,12,13], breast cancer [14,15,16], pancreatic cancer [8,17,18,19,20], lung cancer [21,22,23], gastric cancer [24,25,26,27], cervical cancer [28], uterine serous cancer [29] and cholangiocarcinoma [30,31]. The overexpression of MSLN in triple-negative breast cancer (TNBC) [14,15], ovarian cancer [12], lung adenocarcinoma [21,22], cholangiocarcinoma [31] and pancreatic cancer [17,18] is related to poor prognosis. MSLN is related to chemotherapy resistance, and downregulation of MSLN can restore cell sensitivity to cisplatin in malignant pleural mesothelioma [32,33].

The MSLN precursor is a 71-kDa glycoprotein that is cleaved by enzymes to release the 31-kDa megakaryocyte-enhancing factor (MPF) and 40-kDa mature MSLN. Mature MSLN can generate soluble mesothelin-related peptide (SMRP). The biological effect of SMRP is limited, but it can be quantified by detection in serum and pleural effusion. At present, some clinical studies have identified it as an observation index [33].

The extracellular domain of MSLN consists of region I (N-terminal region; residues 296–390), region II (residues 391–486) and region III (C-terminal region; residues 487–598) (Figure 3). Region I is the membrane-distal region (MDR), which can bind to the mucin MUC16 (also known as CA125). The mucin MUC16 is also expressed in most malignant mesothelioma cells and is associated with tumor aggressiveness. Compared with region I, region III mediates stronger T-cell activation and cytotoxicity and is a better target [34]. The mechanism may be that anti-MSLN CAR-T cells targeting MSLN region I must compete with CA125/MUC16 for the MSLN antigen interaction, which may weaken the binding and function of anti-MSLN CAR-T cells. However, MSLN region III bridges the extracellular domain and transmembrane region of MSLN, which might have a rigid structure or be responsible for a specific function, to provoke a stronger antitumor response [35].

## 2. Clinical Trial Progress of Anti-MSLN CAR-T-Cell Therapy

As of December 2022, a total of 41 clinical trials have been registered on the clinicaltrials.gov website (https://clinicaltrials.gov/). Six studies (NCT02159716, NCT01355965, NCT01897415, NCT02414269, NCT03545815 and NCT01583686) have been published with all but one obtaining clinical outcome events. NCT01583686 did not enter phase II because of the 15 patients enrolled only 1 achieved stable disease (SD) with the remaining patients having progressive disease (PD) or succumbed to their disease [34]. (Table 1).

### 2.1. Pretreatment

Pretreatment with radiotherapy and chemotherapy prior to CAR-T-cell therapy can alter the tumor microenvironment (TME) and host immune response through several potential mechanisms: immunogenic cell death, decreased regulatory T cells (Tregs) [34], localized T-cell infiltration and activation of various proinflammatory factor pathways. Common medications used in preclinical studies include cyclophosphamide, oxaliplatin, fludarabine and albumin-bound paclitaxel before CAR-T-cell therapy [36].

Preclinical studies showed that pretreatment using oxaliplatin combined with cyclophosphamide (Ox/Cy) could improve the migration of CAR-T cells to the tumor, increase the infiltration of CAR-T cells into the tumor and enhance the sensitivity of the tumor to immune checkpoint blockade [37]. The mechanism is as follows: Ox/Cy activates multiple proinflammatory pathways, including the expression of T-cell recruitment chemokines in multiple cells in the TME, which helps CAR-T cells recruit tumors expressing chemokine receptor 5 (CCR5) and C-X-C chemokine receptor type 6 (CXCR6). Upon CXCR3-dependent CAR-T-cell recruitment into tumors, infiltrating CAR-T cells release interferon-γ (IFN-γ), modify the TME to activate M1 macrophages that express the CXCR3 ligands CXCL9 and CXCL10, and then initiate a positive feedback loop.

Cyclophosphamide combined with fludarabine (Cy/Flu) is another common pretreatment used in preclinical studies [38]. Adding fludarabine can improve the proliferation of CAR-T cells and the disease-free survival rate of acute B-cell lymphoblastic leukemia [39]. In neuroblastoma, Cy/Flu-induced lymphoid depletion increases the circulation level of the steady-state cytokine interleukin-15 (IL-15) and increases CAR-T-cell expansion by up to 3 logs [40].

Currently, in clinical trials of anti-MSLN CAR-T cells, most of the conditioning regimens are cyclophosphamide alone (NCT03608618 and NCT02414269) and cyclophosphamide combined with fludarabine (NCT03814447, NCT03799913, NCT01583686 and NCT05531708). One clinical trial (NCT02159716) [41] evaluated the feasibility and safety of anti-MSLN CAR-T cells with and without cyclophosphamide pretreatment in 15 patients, and the results showed that the maximum tolerable dose of anti-MSLN CAR-T cells was 3 × 10^8^ cells/m^2^ and that there were no targeted toxic reactions, such as pleurisy, pericarditis, and peritonitis. However, CAR-T cells did not exert significant clinical efficacy on SD.

### 2.2. Programmed Cell Death Protein-1 (PD-1) and Its Ligand (PD-L1)

There is strong evidence suggesting that the PD-1/PD-L1 interaction between T cells and tumor cells causes the inhibition of T-cell function [42] and depletion of T cells [43]. Therefore, combination with PD-1 or PD-L1 monoclonal antibodies or editing out PD-1 (PD-1 KO) may noticeably enhance the antitumor activity of cytotoxic T lymphocytes (CTLs), enable T cells to recognize tumor cells, promote tumor cell eradication and reduce T-cell exhaustion. Some studies have shown that PD-1KO CTLs are more effective in killing tumor cells than normal CTLs in vitro and in vivo [42,44]. A mouse xenograft tumor model was used to confirm the anticancer activity of PD-1KO CTLs [42]. The therapeutic impact of CAR-T cells may be enhanced by PD-1/PD-L1 inhibition, particularly in the treatment of solid malignancies. Mechanistically, PD-1KO can enhance the antitumor activity of CAR-T cells by blocking PD-1/PD-L1 and PD-1/PD-L2 signaling in these gene-edited T cells.

As a technology to destroy the PD-1/PD-L1 interaction, gene modification has certain advantages in knocking out the PD-1 gene in T cells [42]. The question of whether PD-1 gene editing in host T cells is better than or equally effective to PD-1 mAb treatment is still unanswered due to a lack of solid evidence. However, from recent research results, gene editing of T cells for intracellular intrinsic immune checkpoint blockade may be safer than systemically injecting blocking antibodies into the body [45].

Hu et al. [46] employed CRISPR—Cas9-mediated gene editing to interfere with the PD-1 gene locations in anti-MSLN CAR-T cells to circumvent the inhibitory activity of PD-1 on CAR-T cells. Compared to control CAR-T cells, the release of cytokines (IFN-γ and IL-2) was significantly increased in PD-1 KO CAR-T cells. The results from experiments conducted in vitro and in vivo demonstrated that PD-1 KO CAR-T cells had potent anticancer efficacy against TNBC. The anticancer effect of CAR-T cells was improved in this model by PD-1 KO over PD-1 antibody.

The aforementioned preclinical trials demonstrated the advantages of PD-1 inhibition and CAR-T-cell treatment in combination. To date, the results are very promising and have spurred a growing number of clinical studies. However, a study showed that PD-1 knockout can be problematic, with PD-1 KO T cells exhibiting advantageous short-term proliferation and cytotoxicity, but lacking long-term tolerance and showing vulnerability to T-cell depletion [47]. The therapeutic potential of PD-1 KO or PD-1 antibodies must therefore be completely clarified by additional, carefully conducted follow-up research and clinical studies [47].

In patients with solid tumors, PD-1 inhibition in combination with anti-MSLN CAR-T cells is being studied in terms of both safety and effectiveness. The relevant clinical trials are as follows: NCT05089266, NCT04503980, NCT04489862, NCT03615313, NCT03030001, NCT04577326, NCT03747965, NCT03545815, NCT03182803, NCT02414269 and NCT05373147.

Of all these registered trials, only two have been published. Wang et al. [48] used CRISPR—Cas9 technology to knock down PD-1 and T-cell receptor (TCR) in anti-MSLN CAR-T cells, thereby affecting the TME, in order to treat MSLN-positive solid tumors. A total of 15 patients were included, and CAR-T cells were identified by qPCR in biopsy specimens, which proved that CAR-T cells could effectively infiltrate tumors. The median progression-free survival (PFS) of the seven patients with SD was 7.1 weeks, and most of the eight patients with PD died within 2 months after CAR-T-cell therapy. Adusumilli et al. [49] studied 23 patients who received cyclophosphamide combined with anti-MSLN CAR-T cells followed by at least three doses of pembrolizumab, and the overall survival (OS) following CAR-T-cell therapy was 23.9 months (95% CI, 14.7 months to NE). The one-year OS rate was 83% (95% CI, 68–100%). Among the combined immunotherapy patients (N = 16) with mRECIST measurable disease, two patients (12.5%) had partial response (PR), nine patients (56.3%) had SD, and five patients (31.3%) had PD. The application of combined CAR-T cells and an anti-PD-1 antibody in solid tumors is supported by these data. From this, a phase II study is now underway with a fixed dose of anti-MSLN CAR-T-cell infusion (6×10^7^ CAR-T cells/kg) and pembrolizumab administration four weeks after CAR-T-cell infusion.

After infusion, CAR-T cells can proliferate in the host and further differentiate into memory cells, potentially lasting up to 4 years [50]. However, the presence of these memory cells also increases the likelihood of autoimmune disease. To switch off potential toxicities of PD-1 KO CAR-T-cell therapy, suicide genes can be introduced into the CAR-T structure. Adusumilli et al. [49], Minagawa et al. [51] and Monica et al. [52] linked inducible suicide genes (icaspase9, TK suicide gene) to MSLN-CAR, CD33-CAR, CD44v6-CAR, CD19-CAR and CEA-CAR in their experiments.

### 2.3. Local Administration

The barrier effect of the TME and extracellular matrix (ECM) limits the tumor invasion rate of CAR-T cells. Local administration can cause CAR-T cells to directly enter tumor cells, which greatly improves the tumor infiltration rate.

Mayor et al. and Adusumilli et al. [53,54] showed that intrapleural injection of anti-MSLN CD28 costimulatory (M28z) CAR-T cells eradicated established pleural tumors, even at doses that were lower than intravenous injection of CAR-T cells by 30 times. Furthermore, locally injected CAR-T cells also demonstrated systemic, long-lasting antitumor immunity and were capable of easily traveling from the thoracic cavity to the flanking and peritoneal tumor locations. T-cell imaging revealed that local administration resulted in the accumulation of more CAR-T cells in the tumor at earlier time points.

Several clinical trials (NCT04577326, NCT03608618, NCT03323944, NCT03267173, NCT03198052, NCT03054298, NCT02959151, NCT02706782 and NCT02414269) have been conducted to assess the effectiveness and safety of anti-MSLN CAR-T cells in cancer patients with positive results for MSLN administered via the intrathoracic, intraperitoneal, intratumoral or vascular routes.

## 3. Challenges of Anti-MSLN CAR-T-Cell Therapy

Although anti-MSLN CAR-T-cell therapy offers new hope for patients with solid tumors, many challenges remain. Next, we identify solutions from two aspects: toxicity and technical challenges.

### 3.1. Toxicity

The target antigen MSLN is not specifically expressed on the surface of tumor cells, which may lead to off-target effects. Anti-MSLN CAR-T cells recognize MSLN target antigens and are activated to release cytokines or trigger macrophages to release inflammatory cytokines [55], which may lead to cytokine release syndrome (CRS), neurotoxicity and other adverse reactions.

#### 3.1.1. Off-Target Effects

In a phase I/II clinical trial (NCI-09-C-0041), a patient experienced respiratory discomfort and a decline in blood oxygen saturation within 15 min after completing HER2-CAR-T-cell infusion. Approximately 40 min later, chest X-ray showed pulmonary edema, and he died. It is believed that after the first clearance of HER2-CAR-T cells in the lungs, inflammatory cytokines are subsequently released once the body detects HER2 expressed by healthy lung cells, causing lung toxicity and edema that leads to multiorgan dysfunction and death [56]. MSLN is also expressed in normal mesothelial tissue. While targeting tumor cells, anti-MSLN CAR-T cells may also kill normal tissue cells expressing MSLN, resulting in off-target effects. However, from the results of the five published clinical studies of anti-MSLN CAR-T-cell therapy, no obvious off-target effects have been observed [41,48,49,57,58]. Of course, this may be related to the small number of patients, and it remains to be further investigated. In addition, researchers have developed a variety of methods to reduce off-target effects: (1) The construction of bispecific antibodies [59]. (2) Trans-signaling CARs, which means that the two CARs target different TAAs. CAR1 contains only the CD3ζ signaling domain, and CAR2 contains only the CD28 or other costimulatory factor signaling domain. They are transduced to construct T cells that coexpress two CARs. These CAR-T cells only target cells expressing both TAAs [60,61,62,63]. (3) T cells designed using the synthetic Notch (synNotch) receptor, a new class of synthetic receptors based on the Notch receptor. Antigen A is on the synNotch receptor, and antigen B is on the CAR. Only cells expressing both antigens can be targeted [63,64,65]. (4) Simultaneous introduction of two CARs, one targeting TAA and one targeting inhibitory receptors for antigens present on normal cells rather than tumor cells. CAR-T cells express inhibitory signals when they bind to normal cells and have tumor-killing effects when they bind to tumor cells [63,66]. The above strategies can also be considered in the anti-MSLN CAR-T cell process in the future.

#### 3.1.2. Cytokine Release Syndrome (CRS)

CRS is one of the serious adverse reactions of CAR-T-cell therapy. At present, most studies on CRS are focused on hematological malignancies. Fever, weariness, headache, rash, joint discomfort and myalgia are some of the milder signs of CRS. Hypotension and a high fever are serious symptoms that can worsen and lead to circulatory shock, vascular leakage, disseminated intravascular coagulation and multiple organ failure [67]. An anti-IL-6 receptor antibody (tocilizumab) and symptomatic support therapy (high-dose steroids, vasopressors, ventilatory support, etc.) are all included in the treatment [68]. Some studies suggest that blocking IL-1 may be a new method for treating CRS [69].

The pathogenesis of CRS remains unclear. It has been reported [70] that CAR-T cells release granzyme B, activate caspase 3 and cleave GSDME in target tumor cells, leading to pyroptosis, thereby activating caspase 1 and GSDMD in macrophages and triggering CRS. Elevated GSDME levels in cancer patients were positively correlated with CRS severity.

The type of therapy, the underlying condition and the features of the patient all impact the risk of CRS [67]. The degree of T-cell growth and T-cell activation were linked with CRS severity [71]. The nature of the CAR structure affects the clinical presentation, severity and occurrence time of CRS [72]. The incidence rates of CRS in CAR-T-cell therapies containing CD28 and 4-1BB were 93% and 57%, respectively [73,74]. The development of CRS is impacted by lymphoid depletion before CAR-T-cell injection. Following cyclophosphamide or fludarabine lymphodepletion, the likelihood of developing CRS increases [75]. This may be a result of the increased rate of CAR-T-cell proliferation due to the more significant lymphoid depletion achieved by the combination therapy [67].

Fortunately, no severe CRS has occurred in any published clinical trials of anti-MSLN CAR-T-cell therapy [41,48,49,57,58], which may be related to the small number of published clinical trials. The observations of follow-up clinical trials are highly anticipated.

#### 3.1.3. Neurotoxicity

Similar to CRS, mild to severe neurological dysfunction within days and weeks following CAR-T-cell infusion is termed CAR-T-cell-induced neurotoxicity, which often includes epileptic activity as well as specific deficits such as aphasia, altered eyesight, shaking and facial drooping. Symptomatic support therapy is the main treatment, except for the prophylactic use of levetiracetam during CAR-T-cell infusion, and data on other interventions are limited [76].

At present, little is known about the mechanism of neurotoxicity. According to some studies [77], inflammatory mediators released by macrophages trigger the release of von Willebrand factor and angiopoietin-2 from the Weibel–Palade bodies of endothelial cells in the central nervous system. This replacement of angiopoietin-1 results in the inhibition of TIE receptor tyrosine kinase signal transduction. The integrity of the blood-brain barrier is compromised by endothelial cells, which also become more porous. Coagulopathy is caused by high-molecular-weight von Willebrand factor. As cytokines and activated inflammatory cells continue to cross the blood-brain barrier, this positive feedback loop continues. Because of the pathophysiology’s resemblance to thrombocytopenic purpura, plasmapheresis is being studied as a treatment for neurotoxicity. It has also been postulated [78] that the mechanism may involve NK-cell populations. NK cells secrete IL-2 and IL-15, both of which were found to be elevated in patients with neurotoxicity. This high level of NK cell activation triggers the activation of microglia and mediates a strong pathogenic inflammatory environment in the central nervous system. Fortunately, no serious neurotoxicity has been observed in published clinical trials of anti-MSLN CAR-T-cell therapy [41,48,49,57,58]. Of course, the results of more clinical studies are awaited.

#### 3.1.4. Human Anti-Mouse Antibody (HAMA) Immune Response

The chimeric antibody substitutes the mouse constant region with the constant region of the human antibody-producing gene, which greatly reduces the immunogenic reaction produced by the mouse-derived antibody so that 70% of the antibody components are human components. The murine gene is still present in the chimeric antibody, even though it only makes up a very minor portion of the antigen-recognition sequence of the variable region of the antibody. Clinical trials have shown that chimeric antibodies can also generate HAMA immune responses when applied. The therapeutic efficacy of murine antibodies in humans are constrained by their immunogenicity. The use of murine-derived CAR limits the persistence of CAR-T cells in humans and raises the possibility of allergic responses. The construction of a fully human scFv CAR is one solution.

A clinical trial (NCT01355965) conducted by Maus et al. [79] included a total of four patients who received multiple intravenous infusions of MSLN-targeted mRNA transiently transduced second-generation CAR-T cells. One patient developed anaphylaxis during treatment; he received anti-MSLN CAR-T-cell infusions on days 0, 7 and 49, and anaphylaxis occurred within minutes of infusion on day 49. After exclusion, anti-MSLN CAR-T cells most likely triggered allergic responses by inducing murine antibody sequence-specific IgE antibodies present in CAR-T cells. It is suggested that a single infusion of CAR-T cells is sufficient to achieve efficacy. In this case, with continuous exposure to the product, CAR-T-cell infusion would not induce IgE antibodies.

At present, the results of two clinical trials (NCT02414269 [49] and NCT03545815 [48]) of fully human scFv anti-MSLN CAR-T cells have been published, and no new allergic reactions have been found.

### 3.2. Technical Barriers

The technology related to CAR-T-cell therapy has been continuously improved, but there are still many obstacles that limit its further clinical promotion and application. In the following subsections, we introduce and summarize the corresponding solutions proposed by relevant basic research from the aspects of an immunosuppressive TME, insufficient transport into the tumor, target antigen heterogeneity, proliferation and persistence.

#### 3.2.1. Immunosuppressive TME

The glycolytic metabolism of tumor cells induces hypoxia within the TME, and the accumulation of metabolic waste leads to a decrease in pH and low nutrient content, resulting in oxidative stress [80]. The TME can upregulate immune checkpoint molecules (e.g., PD-1, CTLA-4, TIM3 and LAG3), thereby limiting T-cell function. The TME contains a large number of stromal cells, such as cancer-associated fibroblasts (CAFs) and immunosuppressive cells, including myeloid-derived suppressor cells (MDSCs) [81], tumor-associated macrophages (TAMs), tumor-associated neutrophils (TANs), mast cells and Tregs [82]. M2 TAMs and Tregs can produce transforming growth factor-β (TGF-β) [83] and prostaglandin E2 (PGE2) [84] to protect tumors from immune surveillance and attack. CAR-T cells are sensitive to immunosuppressive mechanisms in the TME.

An immune checkpoint molecular blockade can enhance the antitumor activity of CAR-T cells and is expected to enhance their functional persistence in solid tumors. At present, clinical trials mainly involve knockout of PD-1 by CRISPR–Cas9, PD-1 antibody and anti-PD-1 nanoantibody to combat this immunosuppressive mechanism. Preclinical studies have reported improvement in the activity of anti-MSLN CAR-T cells by knocking down TIM3 via shRNA [85] or by applying TIM3 immune checkpoint inhibitors [86]. TIM3 blockade combined with anti-MSLN CAR-T cells significantly improved their killing potency, cytokine secretion and proliferation. In addition, simultaneous downregulation of the inhibitory receptors PD-1, TIM-3 and LAG-3 on CAR-T cells can enhance the antitumor ability of anti-HER-2 CAR-T cells by upregulating the expression of CD56 [87]. However, this approach has not been applied in anti-MSLN CAR-T cells.

In addition to immune checkpoint inhibitors, oncolytic adenoviruses (OAds) are another immunotherapeutic approach for the treatment of solid tumors. The combination of OAd therapy and CAR-T-cell therapy is a new development direction. OAds expressing tumor necrosis factor-α (TNF-α) and IL-2 enhance and maintain T-cell function, promote the infiltration of anti-MSLN CAR-T cells in tumors, overcome the heterogeneity of tumor target antigen expression and reduce tumor immunosuppression, thereby improving the efficacy of anti-MSLN CAR-T cells in pancreatic cancer [88]. In addition, a study [89] evaluated the combination of anti-MSLN CAR-T-cell and OAd therapies in a TNBC model. It was found that OAds targeting TGF-β could directly lyse tumor cells, with an obvious antitumor response in the early stage and weakened antitumor activity in the later stage. However, anti-MSLN CAR-T-cell therapy has a sustained antitumor effect, with a stronger antitumor response detectable in the later stage. The combination of the two treatments produced a stronger antitumor response.

CD40 is mainly expressed on APCs, including dendritic cells (DCs) and macrophages. CD40L expressed on CD4+ T cells plays a key role in the immune response of DCs and activation of antitumor CD8+ T cells [90]. Studies have demonstrated that CD40 is expressed on activated CD8+ T cells and that CD40+CD8+ T cells can communicate with CD4 through CD40. The direct interaction between CD4+ T cells and CD8+ T cells can encourage CD8+ T-cell cytokine release and cell proliferation [91]. CAR-T cells targeting MSLN region III (MSLN3) were designed to secrete anti-CD40 antibodies. The ratio of cytokines and central memory T cells (TCM) secreted by MSLN3 CD40 CAR-T cells was greater than that secreted by MSLN3 CAR-T cells. MSLN3 CD40 CAR-T cells elicited a stronger antitumor response in vitro and in vivo [92].

To avoid immunosuppressive effects, genetically modifying the CAR can help increase the resistance of T cells to immunosuppression. The introduction of costimulatory molecules and inhibitory cytokines into the CAR can help T cells develop stronger resistance to Tregs, TGF-β and other related immunosuppressive molecules [93,94]. Knockdown of TGF-β receptor II (TGFBR2) by CRISPR–Cas9 technology enabled anti-MSLN CAR-T cells to withstand the negative effects of TGF-β signaling [95]. Adenosine and PGE2 can inhibit the immune system by triggering protein kinase A (PKA). Targeting the adenosine 2A receptor (A2AR) can diminish the inhibitory action of adenosine in vitro. Blocking the localization of PKA to the immune synapse increased the migration of anti-MSLN CAR-T cells to the tumor and enhanced the antitumor effect in a mouse model of melanoma [96]. Additionally, knockdown of A2AR by shRNA resulted in enhanced proliferation, cytokine production and cytotoxic function of anti-MSLN CAR-T cells in a simulated TME [97]. In addition, CAR-T cells can be further modified to express cytokines such as IL-2, IL-12, TNF-α and IFN-γ to escape immunosuppression [98,99].

#### 3.2.2. Insufficient Trafficking into the Tumor

Tumor cells and CAFs form the ECM, which is essential for the progression of cancer. The physical barrier that prevents certain anticancer medicines from penetrating tumor cells is represented by the ECM. In addition, ECM collagen fibers around the tumor restrict T cells from entering the tumor [100,101]. One method to increase the effectiveness of CAR-T-cell treatment is to use matrix degraders. CAR-T cells were engineered to express heparanase (HPSE), which can degrade heparan sulfate proteoglycans, allow CAR-T cells to better infiltrate tumors and increase antitumor activity in mouse models [102]. Another strategy is to exploit the ability of macrophages to secrete matrix metalloproteinases (MMPs) to remodel the ECM so that CAR-T cells can infiltrate tumors [103]. The ECM contains hyaluronic acid (HA), which is broken down by hyaluronidase. The membrane protein PH20, which is naturally expressed by human sperm, has high hyaluronidase activity. Because the PH20 protein has a brief half-life, the IgG2 Fc fragment was incorporated to stabilize the structure of the protein. CAR-T cells expressing sPH20-IgG2 were constructed and showed a strong ability to degrade HA and inhibit tumor growth in a mouse model of xenogeneic gastric cancer [104].

The CAR-T-cell trafficking process requires chemokines secreted by tumor cells to interact with chemokine receptors on CAR-T cells. Chemokines are involved not only in leukocyte recruitment, but also in tumor angiogenesis, cell multiplication and metastasis. It is possible to stimulate more chemokine receptors to be expressed on CAR-T cells; for example, anti-MSLN CAR-T cells can be genetically modified to express the chemokine receptor CCR2 combined with a CXCR4 antagonist. The chemokine receptor CCR2b was introduced into anti-MSLN CAR-T cells to enhance the transport of CAR-T cells to the tumor. Functional CCR2b on anti-MSLN CAR-T cells can significantly increase the number of T cells in tumors and improve the antitumor effect in vitro and in vivo [33]. In addition, CCR2b and CCR4 are receptors for serum monocyte chemotactic protein 1 (MCP-1) and are expressed at low levels on activated T cells. Anti-MSLN CCR2b CAR and anti-MSLN CCR4 CAR-T cells have increased migration rates into tumor supernatants expressing high levels of MCP-1 in vitro. In a mouse model of non-small cell lung cancer, anti-MSLN CCR2b CAR-T cells had better tumor tissue infiltration and antitumor function [105].

Local administration allows the drug to be directly injected into the tumor through the barrier, which can alleviate the obstacle of drug transport into the tumor. This was discussed in detail in Section 2.3.

#### 3.2.3. Target Antigen Heterogeneity

In view of the good efficacy of CAR-T-cell therapy in early clinical trials of hematological malignancies, the FDA approved four anti-CD19 CAR-T cells, namely Kymriah (tisagenlecleucel), Yescarta (axicabtagene ciloleucel), Tecartus (brexucabtagene autoleucel) and Breyanzi (lisocatagene maraluecel) [106]. In addition, idecabtagene vicleucel (ide-cel), which targets BCMA, is awaiting FDA approval. However, subsequent follow-up showed that acute B-lymphocytic leukemia (B-ALL) had a high relapse rate after CAR-T-cell therapy [107]. The results of a clinical study of anti-CD19 CAR-T cells (CTL019) revealed that among the 59 patients enrolled, 55 (93%) achieved CR at 1 month. However, during the 12-month follow-up, the recurrence-free survival rate was only 55% [108]. Loss of target antigens, downregulation of target antigen expression and heterogeneous expression of target antigens are the main causes responsible for relapse after CAR-T-cell therapy [109].

Target antigen heterogeneity is mainly manifested in two aspects, time and space, which we call temporal expression heterogeneity and spatial expression heterogeneity. Among them, the temporal expression heterogeneity of target antigens refers to the fact that antigen expression on the surface of tumor cells can change significantly over time, mainly in the form of antigen loss and antigen expression downregulation [110]. The first published phase I trial of CD19-4-1BB-CD3ζ CAR-T cells for B-ALL at Children’s Hospital of Philadelphia (CHOP) (NCT01626495 and NCT01029366) showed that 3 of 27 CR patients (11%) relapsed due to loss of CD19 in leukemia cells [111]. Another clinical trial of CAR-T cells in the treatment of B-ALL (NCT02315612) showed that 8 of 12 patients (67%) who had CR after CD22 CAR-T-cell therapy relapsed, and 7 of these patients had downregulated CD22 expression. Downregulation of CD22 expression occurs at the posttranscriptional level, and CD22 CAR-T cells cannot effectively eliminate cells with downregulated CD22 expression [112]. This has also been demonstrated in glioblastoma [113,114,115], non-Hodgkin lymphoma [116] and multiple myeloma [117]. However, there is no research on the target antigen MSLN thus far.

Heterogeneity in the spatial expression of target antigens refers to the inconsistent expression of antigens on the surface of different tumor cells in the same patient [110,118]. Rabilloud et al. [119] performed single-cell sequencing analysis of leukemia cells from a patient with relapsed B-ALL, found that CD19-negative leukemia cells were present before CAR-T-cell therapy and confirmed that relapse was due to the cloning of these CD19-negative leukemia cells. A phase I clinical study conducted by Haas et al. [41] investigated the safety and activity of anti-MSLN CAR-T cells in patients with malignant pleural mesothelioma, ovarian cancer and pancreatic ductal adenocarcinoma, (NCT02159716) and found that in only 3 of 15 patients the expression of MSLN on tumor cells was >75%. Thus, clinical trials focusing on efficacy are recommended to prospectively screen MSLN expression to reduce recurrence in the future.

By downloading immunohistochemical (IHC) staining data from The Human Protein Atlas website (https://www.proteinatlas.org/ (accessed on 18 August 2022)) to quantitatively analyze the IHC results of MSLN expression in various solid tumors, we found that there were some malignant cells with negative MSLN expression in solid tumors defined as having positive MSLN expression. In different IHC samples, the positive rates of MSLN expression were approximately 0–99%. In addition, we analyzed the single-cell sequencing results of TNBC (GSE75688 and GSE118389), ovarian cancer (GSE118828) and pancreatic cancer (GSE111672) downloaded from the Gene Expression Omnibus (GEO: https://www.ncbi.nlm.nih.gov/gds/ (accessed on 28 September 2022)) database and found that approximately 70–97% of malignant cells had negative MSLN expression (Figure 4). The heterogeneous expression of MSLN significantly affects the efficacy and recurrence rate of solid tumors after treatment with anti-MSLN CAR-T cells [109].

In view of this, a variety of measures against target antigen heterogeneity have been developed. Among them, enhancing the “bystander effect” of tumor cells is an important strategy. The bystander effect suggests that CAR-T cells can induce tumor killing, even though these malignant cells negatively express the target antigen. To dissect the mechanism of the bystander effect, Upadhyay et al. [120] used CRISPR—Cas9 technology to screen and determine the essential role of Fas-FasL in antigen-specific T-cell killing. They discovered that Fas-FasL mediated the off-target bystander killing effect on antigen-negative malignant cells: the killing effect of CD19 CD3 ζ-CD28 CAR-T cells on target cells showed moderate Fas dependence, and Fas-expressing mice were more sensitive to bystander killing effects. An analysis of pretreatment tumor RNA-sequencing data from the ZUMA-1 trial (NCT02348216), which included patients with CAR-T-refractory diffuse large B-cell lymphoma, showed that patients with a durable response to treatment had significantly increased tumor Fas expression. Similarly, an analysis of a diffuse large B-cell lymphoma patient cohort receiving standard treatment in the TCGA database showed that tumor patients with high expression of Fas had a poorer prognosis; in contrast, patients with high expression of Fas treated with CAR-T cells had significantly longer survival times. This mechanism also applies to anti-MSLN CAR-T cells. Klampatsa et al. [121] designed a model in which anti-MSLN CAR-T cells could cure xenografts in mice inoculated with 100% MSLN-positive tumor cells. However, even if the positivity rate of MSLN dropped to 90%, anti-MSLN CAR-T cells could not cure the tumor. Nevertheless, up to 25% of MSLN-negative xenografts could be cured if pretreatment with a nonlymphoid-depleting dose of cyclophosphamide prior to CAR-T-cell therapy induces a bystander effect. Therefore, exploring the application of multidrug combinations and the application of small-molecule Fas signaling modulators may solve the poor curative effect caused by target antigen heterogeneity to a certain extent.

Park et al. [122] introduced a new method for the exogenous introduction of a homogeneous antigen. Oncolytic vaccinia virus (OV19t) encoding CD19t was used to infect various solid tumor cells. Before the virus-infected tumor is lysed, its cell surface produces new CD19. The expression of CD19t in tumors is encouraged by CAR-T-cell-mediated tumor lysis, which can also cause the release of the virus from tumor cells that are on the verge of death. At this point, all tumor cells are loaded with the new target antigen CD19, potentially extending the application of clinically approved anti-CD19 CAR-T cells to the treatment of solid tumors and overcoming the problem of target antigen expression heterogeneity.

In addition, to address target antigen heterogeneity, multiantigen-targeted CAR constructs are currently being developed, thereby reducing the risk of tumor recurrence. There are numerous methods for creating CARs that target several antigens [123]: (1) dual-targeted CAR-T cells, including bicistronic transgenes [124] and cotransduction and tandem single-chain antibodies [125,126,127,128]; (2) coadministered monospecific CAR-T cells [129]; and (3) sequential monospecific CAR-T cells [129] (Figure 5). In addition, some studies have designed aptamer CAR-T-cell technology. The construction of universal CAR-T cells expressing nontumor-associated antigens, such as fluorescein isothiocyanate (FITC) CAR-T cells, targets multiple target antigens on the surface of tumor cells through a mixture of bispecific aptamers [130] (Figure 6). Kailayangiri et al. [131] found that inhibition of enhancer of zeste homolog 2 (EZH2) could induce the surface expression of GD2 in Ewing sarcoma cells, thereby enhancing the killing ability of anti-GD2 CAR-T cells against Ewing sarcoma cells. These studies suggest that our similar mechanism may also apply to anti-MSLN CAR-T cells.

In a tissue microarray of 107 excised pancreatic cancers, Tholey et al. [132] stained tissues with antibodies against MUC1, MSLN or both. At the protein level, they verified that when MUC1 and MSLN were labeled separately and simultaneously, the percentage of tumor cells with a high labeling pattern (2+ or 3+) increased from 56% and 62% to 82%. To increase the effectiveness, infiltration, persistence and proliferation of CAR-T cells in ovarian cancer, Liang et al. [126] created a unique tandem CAR expressing an anti-FOLR1 scFv, an anti-MSLN scFv and two peptide sequences of IL-12. KRAS mutations were shown to be positively associated with MSLN expression in pancreatic cancer and lung cancer, as demonstrated by Fukamachi et al. [133] and Bauss et al. [134]. It is reasonable to infer that the use of bispecific CAR-T cells (targeting both MSLN and MUC1 or targeting both FOLR1 and MSLN) and CAR-T-cell therapy combined with mutated gene targeting drugs could improve the efficacy of anti-MSLN CAR-T-cell therapy. However, these therapies are still in the preclinical stage, and whether they can solve the issue of tumor recurrence caused by the heterogeneity of the target antigen requires further observation.

#### 3.2.4. Proliferation and Persistence

There are four factors that affect the proliferative capacity and persistence of CAR-T cells: preconditioning, the intracellular signaling domain of CAR-T cells, the immunogenicity of CAR-T cells, and T-cell depletion.

The current commonly used conditioning regimens are cyclophosphamide and cyclophosphamide combined with fludarabine. A clinical trial (NCT02159716) [41] demonstrated that preconditioning with cyclophosphamide before CAR-T infusion can increase the proliferation of CAR-T cells, but has little effect on durability. Preclinical and clinical studies conducted by Goto et al. [135] and Pang et al. [136] showed that the secretion of IL-7 and CCL19 promoted the proliferation and persistence of anti-MSLN CAR-T cells in vivo.

CD28 and 4-1BB are commonly used intracellular signaling domains of CAR-T cells, but their functions are quite different. CD28 CAR-T cells have a better tumor-killing ability, while 4-1BB can better prolong the persistence of CAR-T cells. The reason may be that CD28 CAR-T cells have higher basal activity of nuclear factor of activated T cells (NFAT) and unique sensitivity to PD-1/PD-L1-mediated checkpoint inhibition, while 4-1BB CAR-T cells have stronger nuclear transcription factor kappa B (NF-κB) activity and are not affected by the PD-1/PD-L1 checkpoint [137,138]. Guedan et al. [139] suggested that a single amino acid residue in CD28 causes T-cell depletion and may hinder the persistence of CD28-based anti-MSLN CAR-T cells. Another preclinical study demonstrated that the conversion of asparagine to phenylalanine enhanced the in vivo persistence of anti-MSLN CAR-T cells containing the CD28 costimulatory domain, thereby enhancing their antitumor efficacy [140].

CAR is a protein that also has immunogenicity, so an immune response against CAR is generated in the body, thereby reducing the persistence of anti-MSLN CAR-T cells. Clinical trials (NCT01355965 [57,79], NCT02159716 [41] and NCT01897415 [58]) have reported the detection of human anti-chimeric antibody (HACA) during CAR-T-cell infusion. Constructing a CAR fully targeting human scFv MSLN can reduce the immunogenicity of anti-MSLN CAR-T cells.

The scFv in the CAR structure is unstable and has an inherent tendency to self-aggregate, which may lead to depletion of anti-MSLN CAR-T cells in vivo, thereby reducing their persistence. Studies have shown [109,133] that replacing the scFv with a fully human V_H_ domain or linking the TCR constant region to the heavy and light chain variable regions of monoclonal antibodies produces synthetic T-cell receptors and antigen receptors (STAR). This approach can reduce T-cell exhaustion and results in better or equal cytotoxicity, proliferation and persistence than CAR-T cells.

At present, for anti-MSLN CAR-T-cell therapy, the above four methods are mainly used to improve proliferation and persistence, which are key factors affecting efficacy. In addition, local administration and induction of T-cell costimulators to improve the structure of CAR contribute as well [34].

## 4. Conclusions

Clinical trials of CAR-T-cell therapy in solid tumors have begun in recent years as a result of the approval of the treatment for hematological malignancies. The present state of anti-MSLN CAR-T-cell treatment in patients with solid tumors was discussed in this study. Its toxicity, including off-target effects, CRS, neurotoxicity and immune response, was analyzed. Finally, technical hurdles that may affect its safety and efficacy, including the immunosuppressive TME, trafficking into tumor tissue, target antigen heterogeneity, proliferation and durability, were defined.

At present, the persistence and efficacy of the intravenous infusion of anti-MSLN CAR-T cells still has much room for development. Novel strategies of combination therapy with immune checkpoint inhibitors and/or chemotherapeutic drug pretreatment have improved the antitumor ability of anti-MSLN CAR-T cells to some extent. The application of topical, fully human anti-MSLN scFv, which has entered clinical trials, is also expected to improve the efficacy and reduce the toxicity of anti-MSLN CAR-T-cell therapy. The next steps in the development of new CAR-T-cell therapies for solid tumors will involve multidisciplinary collaboration, focusing on combination therapy and new clinical study designs, and anti-MSLN CAR-T-cell therapy is expected to impact clinical outcomes in patients with various solid tumors.

## Figures and Tables

**Figure 1 cancers-15-01357-f001:**
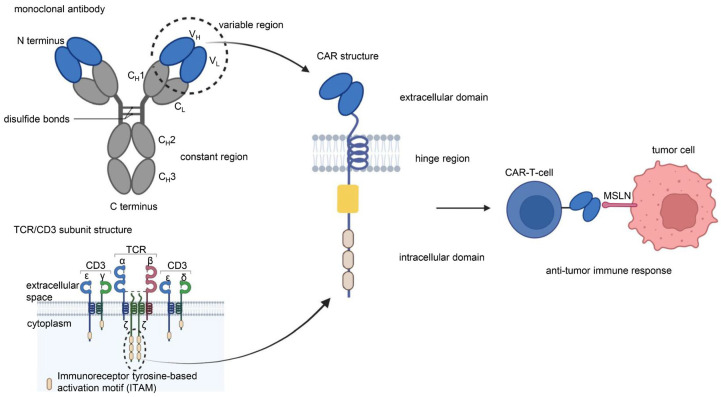
Chimeric antigen receptor (CAR) is a protein composed of three parts: an extracellular antigen-binding domain, an intracellular signaling domain, and a hinge region. The single-chain variable region (scFv) of the extracellular antigen-binding domain of CAR-T cells originates from the V_H_ and V_L_ of monoclonal antibodies. The intracellular signaling domain consists of a costimulatory factor and CD3 signal domain. The CD3 signal domain originates from the immunoreceptor tyrosine-based activation motif (ITAM) of the TCR/CD3 subunit structure. The scFv of anti-MSLN CAR-T cells binds specifically to MSLN on the surface of tumor cells to induce an antitumor immune response. Illustration was created with Biorender.com.

**Figure 2 cancers-15-01357-f002:**
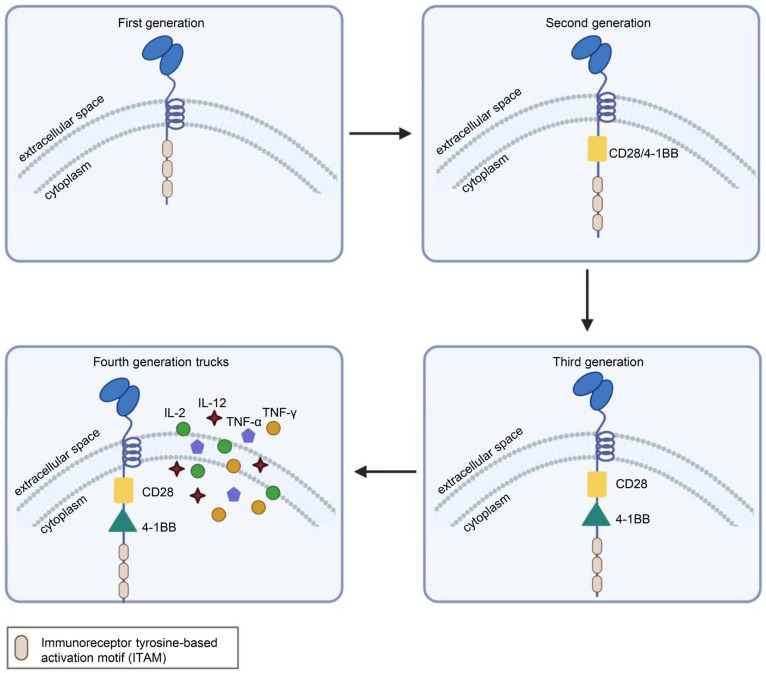
Structural differences in the four generations of CARs. First-generation CARs had only CD3ζ as the intracellular signaling domain. Second-generation CARs contain CD3ζ and a costimulatory domain, which may be CD27, CD28, CD137 (4-1BB) or OX40. Third-generation CARs contain CD3ζ and two costimulatory domains, such as CD28 and CD137 (4-1BB), or other costimulatory molecules. On the basis of the third generation, the latest fourth-generation CARs can additionally secrete cytokines or other effector molecules, such as IL-12, IL-15, IL-7, CCL19 or αPD-1, to modulate the tumor immune microenvironment. Illustration was created with Biorender.com.

**Figure 3 cancers-15-01357-f003:**
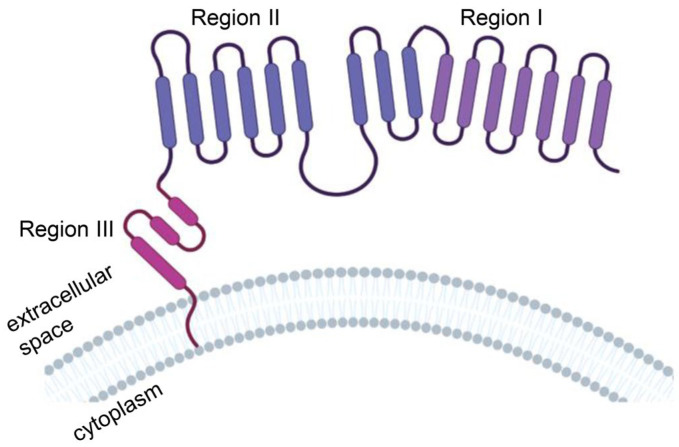
Protein structure model of human mesothelin (MSLN). MSLN is a membrane-binding protein anchored to the cell membrane by glycosylated phosphatidylinositol. The extracellular domain of MSLN consists of region I (N-terminal region; residues 296–390), region II (residues 391–486) and region III (C-terminal region; residues 487–598). Illustration was created with Biorender.com.

**Figure 4 cancers-15-01357-f004:**
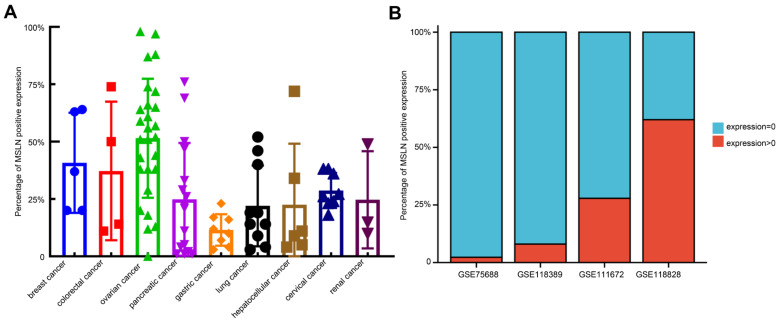
Heterogeneous expression of MSLN in solid tumors. (**A**) Calculation of the expression rate of MSLN in malignant cells of different solid tumors at the protein level with ImageJ. The percentage of MSLN-positive tumor cells among the total number of tumor cells was calculated by IHC. The result of IHC verified by two pathologists. (**B**) Calculation of the expression rate of MSLN in malignant cells of triple-negative breast cancer (GSE75688 and GSE118389), pancreatic cancer (GSE111672) and ovarian cancer (GSE118828) using single-cell sequencing data downloaded from the GEO database (https://www.ncbi.nlm.nih.gov/geo/ (accessed on 28 September 2022)). The percentage of tumor cells with MSLN expression >0 of the total number of tumor cells in the single-cell data was calculated.

**Figure 5 cancers-15-01357-f005:**
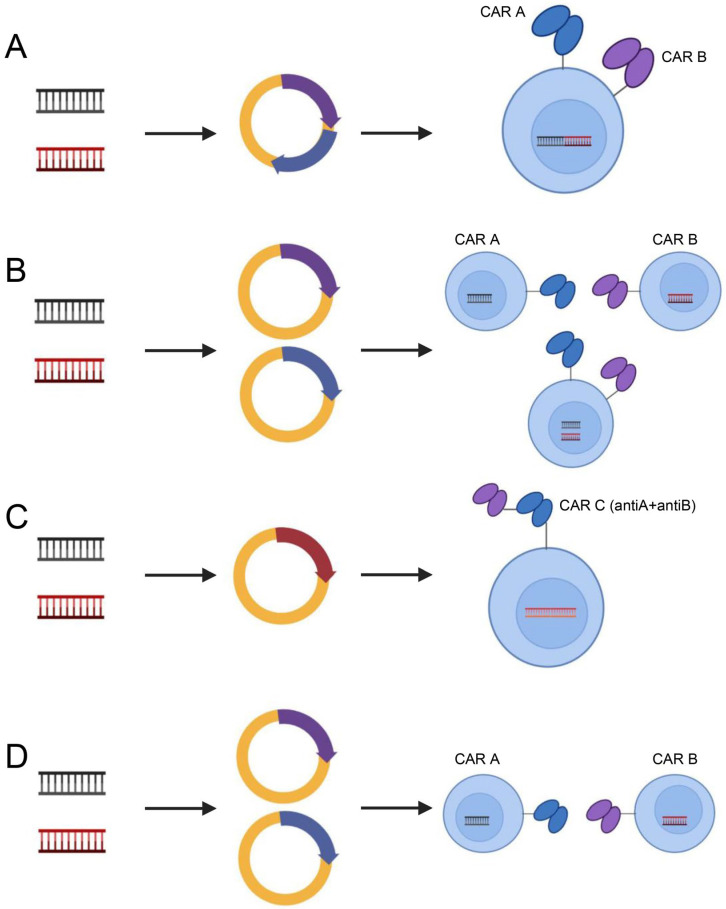
Dual-target CAR-T-cell therapy strategies. Dual-targeting CAR-T cells, including bicistronic transgenic (**A**), cotransduced (**B**), tandem single-chain antibodies (**C**) and coadministered or sequentially administered monospecific CAR T cells (**D**). Illustration was created with Biorender.com.

**Figure 6 cancers-15-01357-f006:**
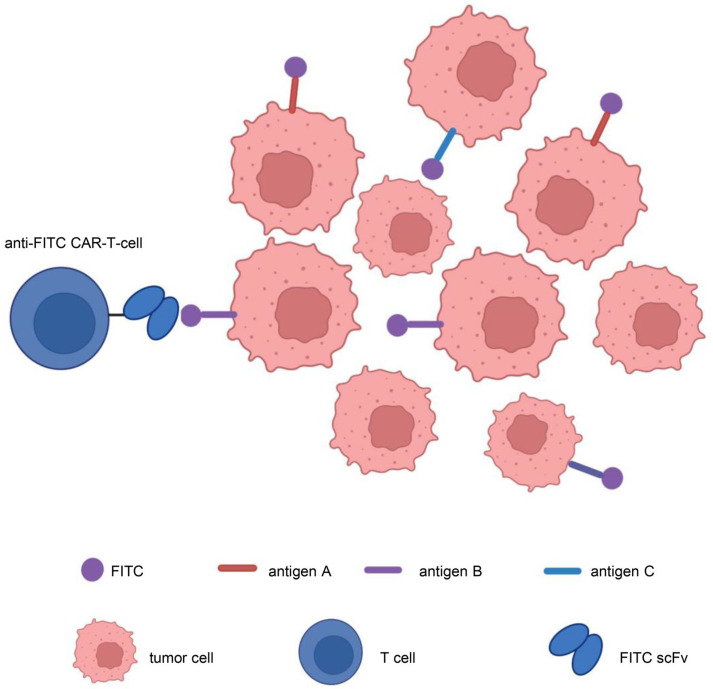
Aptamer CAR-T-cell technology. Construction of universal CAR-T cells expressing nontumor-associated antigens, such as fluorescein isothiocyanate (FITC)-CAR-T cells, targeting multiple target antigens on the surface of tumor cells through a mixture of bispecific aptamers. Illustration was created with Biorender.com.

**Table 1 cancers-15-01357-t001:** Clinical trials using anti-MSLN CAR-T therapy for solid tumors.

NCT Number	Title	Status	Type of Cancer	Phase	Enrollment	Start Date	Publication
NCT05531708	Exploratory Study of Novel MSLN CAR-T Cell Therapy in Patients With MSLN-positive Advanced Refractory Solid Tumors	Recruiting	Refractory Solid Tumors	Phase 1	20	8 September 2022	
NCT05373147	αPD1-MSLN-CAR T Cells for the Treatment of MSLN-positive Advanced Solid Tumors	Recruiting	Solid Tumor	Early Phase 1	21	13 May 2022	
NCT05623488	CAR T Cells in Mesothelin-Expressing Breast Cancer	Not yet recruiting	Breast Cancer	Phase 1	12	21 November 2022	
NCT05089266	Study of αPD1-MSLN-CAR-T Cells to Evaluate the Safety, Tolerability, and Effectiveness for Patients with MSLN-Positive Advanced Solid Tumors	Not yet recruiting	Colorectal Cancer	Phase 1	30	30 November 2021	
NCT05057715	huCART-meso + VCN-01 in Pancreatic and Ovarian Cancer	Not yet recruiting	Pancreatic Cancer, Serous Ovarian Cancer	Phase 1	12	1 December 2021	
NCT04981691	Anti-Mesothelin CAR-T Cells with Advanced Refractory Solid Tumors	Recruiting	Refractory Malignant Solid Neoplasm	Phase 1	12	1 October 2021	
NCT04577326	Mesothelin-Targeted CAR-T-cell Therapy in Patients with Mesothelioma	Recruiting	Malignant Pleural Mesothelioma (MPM)	Phase 1	30	30 September 2020	
NCT04562298	A Phase I Clinical Study to Evaluate the Safety, Tolerability, and Efficacy of LCAR-M23, a CAR-T-Cell Therapy Targeting MSLN in Patients with Relapsed and Refractory Epithelial Ovarian Cancer	Recruiting	Epithelial Ovarian Cancer	Phase 1	34	21 October 2020	
NCT04503980	αPD1-MSLN-CAR-T Cells for the Treatment of MSLN-Positive Advanced Solid Tumors	Recruiting	Colorectal Cancer, Ovarian Cancer	Early Phase 1	10	26 March 2020	
NCT04489862	αPD1-MSLN-CAR-T Cells for the Treatment of MSLN-Positive Advanced Solid Tumors	Recruiting	Non-Small Cell Lung Cancer, Mesothelioma	Early Phase 1	10	13 May 2020	
NCT04203459	The Mechanism of Enhancing the Antitumor Effects of CAR-T on PC by Gut Microbiota Regulation	Recruiting	Pancreatic Cancer		80	20 October 2019	
NCT03941626	Autologous CAR-T/TCR-T-Cell Immunotherapy for Solid Malignancies	Recruiting	Esophageal Cancer, Hepatoma, Glioma, Gastric Cancer	Phase 1 Phase 2	50	1 September 2019	
NCT03916679	MESO-CAR-T-Cell Therapy Relapsed and Refractory Epithelial Ovarian Cancer	Recruiting	Ovarian Cancer	Phase 1 Phase 2	20	20 April 2019	
NCT03814447	Fourth-Generation CAR-T-Cell Therapy for Refractory-Relapsed Ovarian Cancer	Recruiting	Ovarian Cancer	Early Phase 1	10	16 August 2019	
NCT03799913	meso-CAR-T-Cell Therapy Relapsed and Refractory Ovarian Cancer	Recruiting	Ovarian Cancer	Early Phase 1	20	10 April 2019	
NCT03747965	Study of PD-1 Gene-Knocked Out Mesothelin-Directed CAR-T Cells with the Conditioning of PC in Mesothelin-Positive Multiple Solid Tumors	Unknown status	Solid Tumor	Phase 1	10	1 November 2018	
NCT03638193	Study of Autologous T Cells in Patients with Metastatic Pancreatic Cancer	Recruiting	Pancreatic Cancer	Not Applicable	10	11 July 2018	
NCT03615313	PD-1 Antibody-Expressing Meso-CAR-T Cells for Mesothelin-Positive Advanced Solid Tumor	Unknown status	Advanced Solid Tumor	Phase 1 Phase 2	50	6 August 2018	
NCT03608618	Intraperitoneal MCY-M11 (Mesothelin-Targeting CAR) for Treatment of Advanced Ovarian Cancer and Peritoneal Mesothelioma	Terminated	Peritoneal Mesothelioma, Fallopian Tube Adenocarcinoma, Adenocarcinoma of the Ovary, Primary Peritoneal Carcinoma	Phase 1	14	27 August 2018	
NCT03545815	Study of CRISPR—Cas9-Mediated PD-1 and TCR Gene-Knocked Out Mesothelin-Directed CAR-T Cells in Patients with Mesothelin-Positive Multiple Solid Tumors	Recruiting	Solid Tumor	Phase 1	10	19 March 2018	PMID: 34381179
NCT03497819	Autologous CART-meso/19 Against Pancreatic Cancer	Unknown status	Pancreatic Cancer	Early Phase 1	10	1 October 2017	
NCT03356808	Antigen-Specific T Cells Against Lung Cancer	Unknown status	Lung Cancer	Phase 1 Phase 2	20	15 December 2017	
NCT03356795	Intervention of CAR-T Against Cervical Cancer	Unknown status	Cervical Cancer	Phase 1 Phase 2	20	15 November 2017	
NCT03323944	CAR-T-Cell Immunotherapy for Pancreatic Cancer	Active, not recruiting	Pancreatic Cancer	Phase 1	8	15 September 2017	
NCT03267173	Evaluation of the Safety and Efficacy of CAR-T in the Treatment of Pancreatic Cancer	Unknown status	Pancreatic Cancer	Early Phase 1	10	15 June 2017	
NCT03198052	HER2/Mesothelin/Lewis-Y/PSCA/MUC1/GPC3/AXL/EGFR/B7-H3/Claudin18.2-CAR-T Cell Immunotherapy against Cancers	Recruiting	Lung Cancer	Phase 1	30	1 July 2017	
NCT03182803	CTLA-4 and PD-1 Antibodies Expressing Mesothelin-CAR-T Cells for Mesothelin-Positive Advanced Solid Tumor	Unknown status	Advanced Solid Tumor	Phase 1 Phase 2	40	7 June 2017	
NCT03054298	CAR-T Cells in Mesothelin-Expressing Cancers	Recruiting	Lung Adenocarcinoma, Ovarian Cancer, Peritoneal Carcinoma, Fallopian Tube Cancer, Pleural Mesothelioma, Peritoneal Mesothelioma	Phase 1	27	1 March 2017	
NCT03030001	PD-1 Antibody-Expressing CAR-T Cells for Mesothelin-Positive Advanced Malignancies	Unknown status	Solid Tumor	Phase 1 Phase 2	40	15 February 2017	
NCT02959151	A Study of Chimeric Antigen Receptor T Cells Combined with Interventional Therapy in Advanced Liver Malignancy	Unknown status	Hepatocellular carcinoma, Metastatic Pancreatic Cancer, Metastatic Colorectal Cancer	Phase 1 Phase 2	20	1 July 2016	
NCT02930993	Anti-Mesothelin CAR-T Cells for Patients with Recurrent or Metastatic Malignant Tumors	Unknown status	Mesothelin-Positive Tumors	Phase 1	20	1 August 2016	
NCT02792114	T-Cell Therapy for Advanced Breast Cancer	Active, not recruiting	Metastatic HER2-Negative Breast Cancer	Phase 1	186	1 June 2016	
NCT02706782	A Study of Mesothelin Redirected Autologous T Cells for Advanced Pancreatic Carcinoma	Unknown status	Pancreatic Cancer	Phase 1	30	1 March 2016	
NCT02580747	Treatment of Relapsed and/or Chemotherapy Refractory Advanced Malignancies by CART-meso	Unknown status	Malignant Mesothelioma, Pancreatic Cancer, Ovarian Tumor, Triple-Negative Breast Cancer, Endometrial Cancer, Other Mesothelin-Positive Tumors	Phase 1	20	1 October 2015	
NCT02465983	Pilot Study of Autologous T cells in Patients with Metastatic Pancreatic Cancer	Terminated	Pancreatic Cancer	Phase 1	4	1 May 2015	
NCT02414269	Malignant Pleural Disease Treated with Autologous T Cells Genetically Engineered to Target the Cancer-Cell Surface Antigen Mesothelin	Active, not recruiting	Malignant Pleural Disease, Mesothelioma, Metastases, Lung Cancer, Breast Cancer	Phase 1 Phase 2	113	1 May 2015	PMID: 34266984
NCT02388828	CART-meso Long-Term Follow-up	Completed	Subjects Who Have Received Lentiviral-Based CART-meso Therapy		10	1 March 2015	
NCT02159716	CART-meso in Mesothelin-Expressing Cancers	Completed	Metastatic Pancreatic (Ductal) Adenocarcinoma, Epithelial Ovarian Cancer, Malignant Epithelial Pleural Mesothelioma	Phase 1	19	1 June 2014	PMID: 31420241
NCT01897415	Autologous Redirected RNA Meso-CAR-T Cells for Pancreatic Cancer	Completed	Metastatic Pancreatic Ductal Adenocarcinoma (PDA)	Phase 1	16	1 July 2013	PMID: 29567081
NCT01583686	CAR-T-Cell Receptor Immunotherapy Targeting Mesothelin for Patients with Metastatic Cancer	Terminated	Cervical Cancer, Pancreatic Cancer, Ovarian Cancer, Mesothelioma, Lung Cancer	Phase 1 Phase 2	15	4 May 2012	
NCT01355965	Autologous Redirected RNA Meso-CAR-T Cells	Completed	Malignant Pleural Mesothelioma	Phase 1	18	1 May 2011	PMID: 24579088

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
