# Peer review of "Challenges of Anti-Mesothelin CAR-T-Cell Therapy"

_cancers, 2023, doi:10.3390/cancers15051357_

Round 1

Reviewer 1 Report

The authors provide an extensive review on anti-mesothelin CAR T cell therapy.  Overall, the manuscript is clear and well written. Its only major limitation is that the authors performed their search for clinical trials in November 2021: since then, over a year has passed. Therefore they should update their work, including also clinical trials started after November 2021 and paper published since then.

Other  minor issues should be addressed as well:

- Paraghaph 2.1. "maximum tolerated dose [...] was 3x108 cells/m2. I suppose that the authors mean 3x108 cells/m2. Please check.

- Paragraph 3.1.4: "CAR-T cell infusion will not produce IgE antibodies". I suggest to use instead "would not induce" or "is not expected to induce".

Reviewer 2 Report

This is a well-written paper by Xue-jia Zhai and colleagues aimed at review of current reports on advancements in clinical studies of anti-mesothelin CAR-T-2 cell therapy. The manuscript is clear, properly organized and contain all required information for review paper. The authors extensively discuss recent reports on the clinical research status, obstacles, progress 17 and challenges of anti-MSLN CAR-T cells therapy, and summarizes the relevant programs to improve the efficacy and safety of anti-MSLN CAR-T cells therapy. The idea of this review is focused on presentation of the CAR-T cells technology, MSLN background, clinical trial progress of anti-MSLN CAR-T-cell therapy, adjuvant therapy to anti-MSLN CAR-T-cell therapy, role of programmed cell death protein-1 (PD-1) and its ligand (PD-L1) during therapy, therapy side-effects and limitations. These issues are clearly summerised and illustrated in Table 1 and Figures 1 - 5. Paper has 135 references which are relevant to article's subject.

There is lack of such comprehensive survey focused on anti-MSLN CAR-T cells therapy against different cancers beyond mesothelioma available in the oncology literature. Thus, this is an interesting review, and clinically valuable, especially for those oncology clinicians who seek for anti-MSLN CAR-T cellsas a prospective therapy. This manuscript provides comprehensive information on this modality in anti-cancer therapy.

Taken together, paper by Xue-jia Zhai and colleagues represent a worthwhile contribution to the cancer research. I recommend the manuscript for further publication process.

Reviewer 3 Report

In this article the authors review mesothelin targeting chimeric antigen receptor T-cells and outline the advances, obstacles and challenges of anti-MSLN CAR T-cell therapy. The authors begin by introducing CAR T-cells and mesothelin. They then review current clinical trials using anti-MSLN CAR T-cells before reviewing the challenges currently face in anti-MSLN CAR T-cell therapy.

Overall, I think this review still needs work, particularly sections 1 and 2. This is important as section 1 is introducing the key elements of the review so needs to be more thorough and clear. Section 2 needs the most work. The whole review is based on these clinical trials; therefore, it is important to properly describe them, their findings and place them in context with some of the hurdles you are describing in section 3. Section 3 reads better and covers the concepts in more depth.

I would also question the novelty of this review, as this review by Klampatsa et al 2021 covers the same topic with very similar findings. I would suggest that the authors try to expand on the areas not covered by this already published review to try and adopt a new angle.

Astero Klampatsa, Vivian Dimou & Steven M. Albelda (2021) Mesothelin-targeted CAR-T cell therapy for solid tumors, Expert Opinion on Biological Therapy, 21:4, 473-486, DOI: 10.1080/14712598.2021.1843628

I would like to offer the authors this specific feedback on the review as it is. However, even if all these changes are made I still think the authors need to restructure or put a different emphasis on this review to bring in new ideas not already covered by other reviews in order for this study to be published.

Figure 1. It would be useful if the authors included arrows showing which parts from the antibody and CD3 form the chimeric antigen receptor. They also need to define the orange box. It may be useful to label the individual parts of the CAR for clarity.

Paragraph starting line 64. It would be useful to describe the advantage of having two co-stimulatory domains in the third generation CAR

Line 109 What do the numbers in the brackets mean? Are they amino acid numbers, if so you need to better define. How does region 1 combine with MUC16? Perhaps you mean interact with rather than combine?

Line 112 What makes it a better target? Is it more accessible to binding by scFv or is it a better conformation. Why or how does region III of mesothelin mediate T-cell activation & cytotoxicity? Why would mesothelin elicit cytotoxicity? I think the authors are trying to suggest that mesothelin antibodies or CARs targeting region III elicit stronger T cell activation and cytotoxicity, but this is not clear and should be addressed.

Line 116. Why have the authors only covered clinical trials before November 2021, this needs to be updated to insure there hasn’t been any progress in the last 12 months.

Line 118. This paragraph is very confusing and the point the authors are trying to make is not clear. I assume the authors are trying to provide of the summary of the results from the six completed trials with publications however this is not clear from this paragraph. I would suggest re-wording but also expanding and providing more in depth analysis of the trials and their findings. There is very little discussion of the clinical trials to justify the four pages of clinical trials listed in table 1.

Table 1 is not well formatted and it is not clear which types of cancer belong to which trial. I would suggest spacing out the table or adding lines to separate each trial. It would also look better if the words fit into the columns without running on to the next line.

Line 128 I would suggest that the authors remove the term “etc.“ as it implies there are more immune responses that you have not listed. All potential immune responses should be listed.

Line 128 This sentence is a bit ambiguous.  It is not clear whether the medications are used in clinical trial cancer patients in general, in combination with CAR T-cell therapy or before CAR T-cell therapy.

Line 136 What are partially dependent recombinant CCR5 and CXCR6 expressing tumours? Recombinant suggests artificially generated in the lab so how relevant would this model be to tumors in patients?

Authors need to make it clear that the studies described in the paragraphs starting line 131 and line 142 are pre-clinical studies, not clinical trials in humans.

Line 153 Fix the superscript for the number of CAR T-cells

Line 157 and 158 This sentence doesn’t read well I would suggest “causes inhibition of T-cell function and depletion of T-cells”

Line 182 It’s not clear what the authors mean by aforementioned trials as the authors haven’t mentioned clinical trials combining anti-PD-1 or anti-PD-L1 & CAR T-cells. Perhaps they mean preclinical studies?

Line 184 This sentence is confusing. I would suggest “ However, a study (since the authors only mention one) has shown that PD-1 knockout can be problematic, with PD-1 KO T-cells exhibiting…

Line 209 Fix the superscript

Line 232 I would suggest positive results instead of positivity

The sentence from line 256-266 is very long and needs to be broken down to help improve readability.

Line 262 What is the synNotch receptor? This needs further clarification.

Line 329 This sentence doesn’t read well and I’m not sure what the authors mean by the ability of CAR T-cells to remain in the receptor.

Line 366 Which other ways? It is important that the authors include all strategies to be complete.

Line 390 I would suggest re-wording to “activation of antitumor CD8+ T-cells”

Line 402 I would suggest using the word enabled instead of renders.

Line 433 This example is a bit confusing. How does CCR2 genetically modify a CAR T-cell? Maybe the authors should consider re-wording for clarity.

Paragraph on Line 443 has already been covered in section 2.3 and this paragraph doesn’t add any new information.

Line 549 Perhaps introduce this as epigenetic modification in case readers are unaware what EZH2 inhibitors are.

Figure 4 I would suggest altering the figure legend and combining both parts marked (D) as this is unconventional. Perhaps “ co-administered or sequentially administered monospecific CAR T-cells (D)”

Reviewer 4 Report

The review by Zhai et al. is informative and focused on the MSLN antigen targeting CART cells. Even though this review is written comprehensively and lucidly, here are some suggestions for improving it:   

Figure 1 is a poor representation. The authors have shown a CAR structure, the components of CARs, and the anti-tumour response, but the figure should be improved and the figure legend should explain what is shown in the figure. The sequence of the figures can be shown by drawing arrows or numbering them. It appears that all four figures in the panel are unrelated or independent of one another.

2. Lines 75-80 are redundant and match lines 66-71. These lines in the legend are unnecessary. The 4 generations of CAR have been presented and reviewed in multiple previous reviews. The authors are encouraged to present anti-MLSN CAR structures if possible/generations in clinical trials.  

3. The author should cite the specific reference and cancer type where downregulation of MLSN restores sensitivity to cisplatin (line 103)

4. It would be better to represent lines 109-113 in a figure than just showing the 4 generations of CAR in figure 2. 

5. Language used in the manuscript should be improved and authors must use better words/phrases (e.g., "CART-therapy has achieved good results"); in multiple sections of the manuscript, the language used sounds unprofessional.

· Lines 157- 158 are misleading in that PD-1/PDL1 interaction causes T-cell depletion

·      Line 160, genetic engineering of PD-1 KO

·      Line-169-170, technological means of destroying PD1-PDl1 relationship

·      Line-214-217 need rewriting

6.     Reference is missing for the example given in lines 185-187

Referencing is not consistent throughout the manuscript. References are sometimes cited in the middle of sentences (e.g. line 451).

7.     3.2.3 The section is primarily devoted to CD19, perhaps other antigens could also be discussed or the author could focus solely on MSLN.

8.     Line -495-497, the authors do not mention any references for these lines. Is this the authors' hypothesis based on the data they analyzed?

9.     The phrases used in figure 3 are confusing “Expression rate, Y axis labels- “percentage of positive expression” it doesn’t make any sense.

10. Figure legend is also difficult to follow, and the data analysis methodologies are unclear. 

Round 2

Reviewer 4 Report

The authors have addressed all my major and minor comments, and the review has greatly improved in quality.

I would like to suggest increasing the font size of the axis labels of the Figures to make them more visible without zooming in. Overall, my recommendation is to publish this review article after minor editing.